# Constructing Strategies and Applications of Nitrogen-Rich Energetic Metal–Organic Framework Materials

**Luping Xu [1,\*], Juan Qiao [1], Siyu Xu [1], Xiaoyu Zhao [2], Wanjun Gong [1] and Taizhong Huang [3,\*]** 

1   Xi'an Modern Chemistry Research Institute, Xi'an 710065, China; qiaojuan_2000@163.com (J.Q.); xusy99@163.com (S.X.); xhx204@163.com (W.G.)
2   North Institute of Science and Technology Information, Beijing 100089, China; zhaoxy81@tom.com
3   School of Chemistry and Chemical Engineering, University of Jinan, Jinan 250022, China
\*   Correspondence: sungirlxu5@163.com (L.X.); chm_huangtz@ujn.edu.cn (T.H.);
    Tel.: +86-29-81149923 (L.X.); +86-531-89736103 (T.H.)

**Abstract:** The synthesis of energetic metal–organic frameworks (EMOFs) with one-dimensional, two-dimensional and three-dimensional structures is an effective strategy for developing new-generation high-energy-density and insensitive materials. The basic properties, models, synthetic strategies and applications of EMOF materials with nitrogen-rich energetic groups as ligands are reviewed. In contrast with traditional energetic materials, EMOFs exhibit some interesting characteristics, like tunable structure, diverse pores, high-density, high-detonation heat and so on. The traditional strategies to design EMOF materials with ideal properties are just to change the types and the size of energetic ligands and to select different metal ions. Recently, some new design concepts have come forth to produce more EMOFs materials with excellent properties, by modifying the energetic groups on the ligands and introducing highly energetic anion into skeleton, encapsulating metastable anions, introducing templates and so on. The paper points out that appropriate constructing strategy should be adopted according to the inherent characteristics of different EMOFs, by combining with functional requirements and considering the difficulties and the cost of production. To promote the development and application of EMOF materials, the more accurate and comprehensive synthesis, systematic performance measurement methods, theoretical calculation and structure simulation should be reinforced.

**Keywords:** high-energy-density materials; metal–organic frameworks; nitrogen-rich compounds; constructing strategy

## 1. Introduction

The materials with metal–organic frameworks (MOFs) are kinds of coordination compounds. The concept of the metal–organic framework was first proposed by ProfessorYaghi in the journal *Nature*, in 1995. The MOFs are ordered network structures composed of inorganic metal ions or metal clusters and organic ligands, and they are assembled from the configuration of metal ions and the coordination modes of ligands to form one-dimensional (1D), two-dimensional (2D) and three-dimensional (3D) structures [1] on the basis of coordination chemistry. The MOFs are also called the porous coordination polymers (PCP) [2] for the porous structure. Most of the organic ligands are multidentate organic (such as aromatic polyacids and polybasic) that contain oxygen, nitrogen, etc. The inorganic metals are mostly transition metals, rare earth metals and main group metals, such as manganese(Mn), cobalt(Co), nickel(Ni), zinc(Zn), copper(Cu) and so on [2–6]. Therefore, in terms of the chemical structure, the compounds combine the rigidity of inorganic compounds with the

flexible characteristics of organic polymer materials. Thus, the compounds exhibit many advantages, such as porosity, large specific surface area, high density, perfect thermal stability, structural diversity and designability, adjustability of channel size and so on. They have been used in the various fields of ion exchange, gas storage, catalysis, optics, medicine and fuel cells, among others [7–9]. Over the past 20 years, investigations on the MOFs-based materials have evolved from simple crystal structure research to property exploration, and then gradually evolved into purposefully designed and synthesized target crystals based on functional properties [10–22]. The nitrogen-rich energetic organic groups are introduced into the MOFs to synthesize a novel type of MOFs materials called energetic MOFs (EMOFs) materials [23–28]. It is an efficient method for the advancement of high-energy-density materials with low sensitivity. At the beginning of 20th century, the pioneering scientists, such as Ugryumov, KlapÖtke [29], Shreeve [30] and Cudzilo [31], had successfully synthesized many EMOF materials. These materials show many inspiring characteristics [32–35], such as high energy and enormous heat of detonation, when compared with traditional energetic materials.

## 2. The Family of EMOF Materials

### 2.1. Materials of 1D, 2D and 3D EMOFs

According to the different coordination directions of energetic ligands with metal ions, the energetic metal–organic frameworks can be divided into one-dimensional linear, two-dimensional planar and three-dimensional network. In 2012, HÖpe-Week put forward the concept of EMOFs [36] in a publication and successfully synthesized three 1D EMOFs materials with hydrazine as an energetic ligand, $[Ni(N_2H_4)(ClO_4)_2]_n$ (NHP, N% = 33.49%, $\rho$ =1.983 g·cm$^{-3}$, $T_d$ = 220 °C), $[Co(N_2H_4)_5(ClO_4)_2]_n$ (CHP, N% = 33.49%, $\rho$ = 1.948 g··cm$^{-3}$, $T_d$ = 194 °C) and $[Ni(N_2H_4)_3(NO_3)_2]_n$ (NHN, N% = 40.14%, $\rho$ = 2.156 g·cm$^{-3}$). Some examples are given, as follows, to describe different structures.

In the 1D crystal structure (Figure 1) of $[Cd(en)(N_3)_2]_n$ (N% = 43.67%, $\rho$ = 2.090 g·cm$^{-3}$) [37–40], the Cd(II) ion is six-coordinated in a distorted octahedral geometry with four azide ligands by $\mu$−1,1 azide bridges and two ethylenediamine molecules which serve as bidentate ligands through nitrogen atoms. From the thermal analysis, the decomposition temperature is 149.85°C, and it shows good thermal stability. Under a nitrogen atmosphere with a heating rate of 10 K·min$^{-1}$, the final decomposed residue at 452 °C is Cd. From the sensitivity measurements, the compound has higher friction sensitivity and lower flame sensitivity compared to nickel (II) hydrazine.

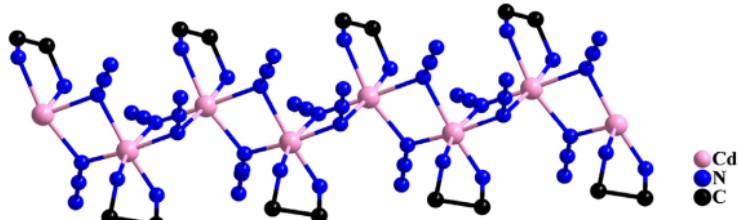

**Figure 1.** The 1D chain structure along a-axis in $[Cd(en)(N_3)_2]_n$ and octahedral coordination structure of Cd(II) [37].

$[Cd(DAT)_2(N_3)_2]_n$ (DAT = 1,5-diaminotetrazole, N% = 63.42%, $\rho$ = 2.144 g·cm$^{-3}$, $T_d$ = 208 °C) is a 2D EMOFs compound with better properties than that of $[Cd(en)(N_3)_2]_n$. In the structure (Figure 2), each Cd(II) atom is six-coordinated with two trans DAT ligands and four trans−1,3 azido bridged ligands [40]. The thermal analysis shows that the compound would produce mainly environmentally friendly gases.

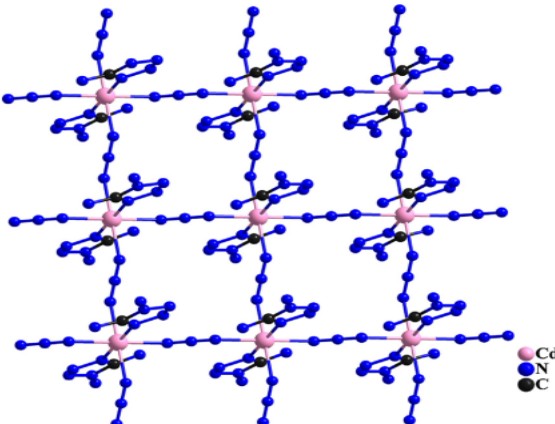

**Figure 2.** The two zigzag chains linked by azido ligands are vertical to each other, and a novel 2D rectangulargrid-like layer parallel to the *ab*-plane of unit cell is formed [37].

$[Zn(C_6H_4N_5)N_3]_n$ [41] with excellent thermal stability up to 345 °C is a nonporous 3D EMOF material. In the structure, the asymmetric unit is composed of a Zn atom, coordinated to both azide and 3-pyridyltetrazole molecules (Figure 3a). The zinc (II) cation shows a distorted trigonalbipyramidal orbital geometry, in which N1, N5 and N6 from tetrazolide, pyridyl and azido ligands, respectively, are displaced in the equatorial plane. N3 and N6 from azido and tetrazolide groups form coordination bonds. Figure 3b shows the presence of azido groups in the axial and equatorial positions leading to a symmetrical arrangement of two trigonalbipyrimidal units related by point symmetry and bonded through azido bridges by a µ−1,1coordination mode. This compound shows enhanced thermal stability because 3-pyridyltetrazole and azido ligands are stabilized in its structure. The measured enthalpy of combustion ($-3623$ kJ·mol$^{-1}$) is higher than trinitrotoluene (TNT), 1,3,5-trinitro-1,3,5-triazacyclohexane (RDX) and 1,3,5,7-tetranitro-1,3,5,7- tetrazocane (HMX),but the measured heat of detonation is moderate, possibly due to the relatively low oxygen content in the structure. The detonation velocity (D) was found to be 5.96 km·s$^{-1}$, and the pressure of detonation (P) is 9.56 GPa.

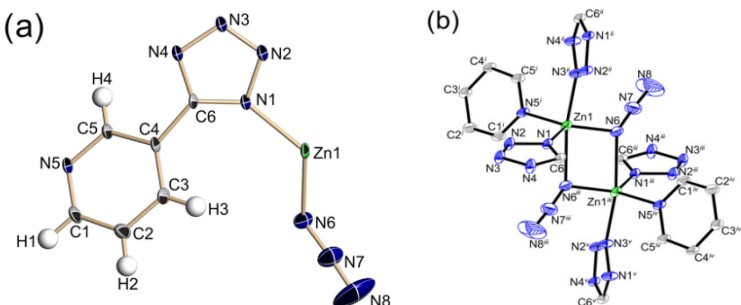

**Figure 3.** (**a**) Thermal ellipsoid plots of the asymmetric unit of compound with the 50% probability level, whereas hydrogen atoms are drawn as spheres of arbitrary radii. (**b**) Thermal ellipsoid plot of symmetrical molecular arrangement. Hydrogen is omitted for clarity [41].

Other 1D, 2D and 3D EMOFs compounds of Ni-DNBT($\rho$ = 1.694 g·cm$^{-3}$, dry, DNBT =5,5′-dinitro−2H,2H′−3,3′-bi−1,2,4-triazole, 1D) [42], Cu-DNBT ($\rho$ = 1.960 g·cm$^{-3}$, dry, 1D) [42], $[Co_2(N_2H_4)_4(N_2H_3CO_2)_2]2ClO_4·H_2O$(CHHP, N% = 23.58%, $\rho$ = 2.000 g·cm$^{-3}$, $T_d$ = 231 °C, 2D), $Zn_2(N_2H_4)_3(N_2H_3CO_2)_2]2ClO_4·H_2O$ (ZnHHP, N% = 23.61%, $\rho$ = 2.117 g·cm$^{-3}$, $T_d$ = 293 °C, 2D) [43], $[Cu(Htztr)]_n$ (2D), $[Cu(Htztr)_2(H_2O)_2]_n$ [44] (H$_2$tztr =3-(1H-tetrazol-5-yl)-1H-trizaole, 3D), ${[Cu(tztr)]·H_2O}_n$ (3D), $[Cu(atrz)_3(NO_3)_2]_n$ (atrz = 3-amino-1,2,4-triazole, 3D), $[Ag(atrz)_{1.5}(NO_3)]_n$ (3D) [45], $[Ag_{16}(BTFOF)_9]_n[2(NH_4)]_n$ [46] (H$_2$BTFOF = 4,4′-oxybis [3,3′-(1H-5-tetrazol)]furazan, 3D), potassium 4,4′-bis(dinitromethyl)-3,3′-azofurazanate(K$_2$DNMAF, 3D) [47] and others also have been reported [48].

## 2.2. Neutral, Cationic and Anion EMOFs

The EMOFs can also be divided into neutral, cationic and anion frameworks based on whether the main body skeleton has electric charges or not.

Neutral EMOFs are formed by coordination between metal ions and deprotonized ligands, and their structural channels often contain small solvent molecules, such as water or $CH_3OH$, to stabilize the skeleton structure [49]. For example, the compounds of $[Cu(bta)(NH_3)_2]_n$, $[Cd(en)(N_3)_2]_n$ with 1D EMOFs, $[Cd(DAT)_2(N_3)_2]_n$, the compounds of $[Pb(Htztr)_2(H_2O)]_n$ [50] with 2D EMOFs, and the other compounds of $[Cu(Htztr)]_n$, $[Cu(tztr)(H_2O)]_n$ with 3D EMOFs all have neutral main body skeletons in the EMOFs [51]. In the crystal structure of $[Cu(tztr)(H_2O)]_n$, each Cu(II) ion coordinates with six nitrogen atoms from $H_2$tztr ligands through sixcoordination modes. The N1 and N2 atoms in the $H_2$tztr ligand adopt chelating modes to connect to one Cu ion, whereas the N3, N4 and N5 atoms adopt monodentate bridging modes to link with three corresponding Cu ions. The molecule of tetrazoliumtriazole loses two hydrogen ions to became a bidentate ligand with two negative charges. Therefore, the main skeleton is neutral, and the water molecules fill in the pores. In the crystal structure of $[Cu(Htztr)]_n$, each copper has a positive charge, and the main skeleton of compound is still neutral, in which there is no water molecule.

Cationic EMOFs are when metal ions coordinate directly with nitrogen atoms with lone pair electrons in the energetic ligands, so the main skeleton has a cationic charge. At this time, there will be corresponding anions in the structure channel to balance the charge of the main skeleton. Common inorganic anions would be $NO_3^-$ and $ClO_4^-$ [49]. Due to the low heat of formation, the inorganic anion shaves little significance to improve the energy performance of EMOFs materials. Meanwhile, $ClO_4^-$ anions would transform into environmentally harmful chlorides (such as HCl) when the materials are burned or exploded. Therefore, the energetic anion used to replace the inorganic anion to balance the charge of the skeleton would be a research focus. For example, the research team of Beijing institute of technology of China has developed ten kinds of EMOF materials containing polynitroazole ring anions [38], such as 2D $\{Cu(atrz)_2(C_2N_5O_4)_2 2H_2O\}_n$ (MOFs(DNT)), $\{Cu(atrz)_2(C_3N_5O_6)_2 2H_2O\}_n$ (MOF(TNP)), $\{Cu(atrz)_2(CN_5O_2)_2 2H_2O\}_n$ (MOF(NTT)), etc. Among them, 3,5-dinitro-1,2, 4-triazole anions ($C_2N_5O_4$), 3,4,5-trinitropyrazole anions ($C_3N_5O_6$) and 5-nitrotetrazole anions ($CN_5O_2$) lie in the 2D planar skeleton. For example, the schematic diagram of the layered structure of MOFs (DNT) is given in Figure 4. In addition, nitrogen and oxygen bonds are introduced into the nitrozole rings, to synthesize the corresponding EMOF materials, such as 2D EMOFs(DNTO) and EMOFs(TNPO) and 3D EMOFs(NTTO). Due to the introduction of the N-O bond, the structure density would increase, so the material density and oxygen balance could be improved, and the energy level could be further improved.

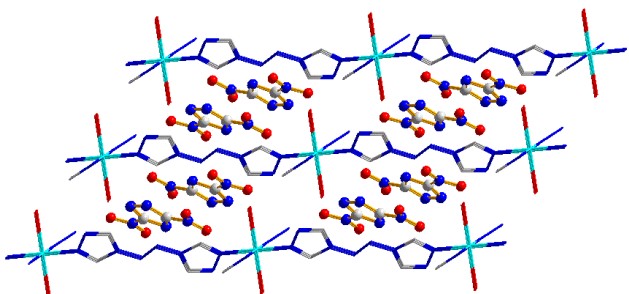

**Figure 4.** Structure of metal–organic frameworks (MOFs, DNT) with anion in the frameworks [38].

Anionic EMOFs means that when metal ions coordinate directly with nitrogen atoms with lone pair electrons, the main body skeleton has an anion charge, and corresponding cations exist inside the structure in order to balance the charge of the main body skeleton. Compared with the first two types of EMOFs, the preparation of anionic EMOFs material is more difficult, because cationic skeleton always appears in the traditional preparation process, and the complex preparation process would

increase the difficulty of controlling the synthesis process and often leads to risks [52]. For all of this, many EMOFs materials with anionic skeleton have been successfully synthesized. These anions are given in Figure 5 [53].

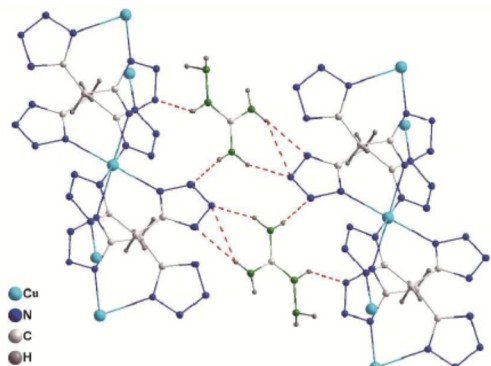

**Figure 5.** Some cations to balance charges of energetic metal–organic frameworks (EMOFs) [53].

These cations are hydrogen donors and have rich nitrogen groups, which could increase the nitrogen content and form multiple strong hydrogen bonds in the skeleton, to make the skeleton more stable, so as to achieve the energetic materials with highenergy, highstability and lowsensitivity that have been pursued by researchers [49,54].

In 2016, the research group of Beijing institute of technology reported that two cases of anionic EMOF materials with bitetrazomethane (BTM) as ligand and $AG^+$ as template were successfully synthesized: $[(AG)_3(Co(BTM)_3)]$ and $\{[AG]_2(Cu(BTM)_2)]\}_n$ [49]. The asymmetric unit of $\{[AG]_2(Cu(btm)_2)]\}_n$ consists of one divalent Cu(II) ion, two BTM dianions and two aminoguanidinium cations. In the structure (Figure 6), each copper ion coordinates with six nitrogen atoms from four BTM, and BTM has a three-coordination mode. The copper ions and the BTM ligands form a two-dimensional layered structure with infinite extension, and $AG^+$ lies between adjacent layers, leading to a large number of hydrogen bonds, by which the 2D coordination polymer finally forms a 3D network. The performance characteristics of two EMOF materials are given as follows: $[(AG)_3(Co(btm)_3)]$(N% = 68.7%, $\rho$ = 1.684 g·cm$^{-3}$, $T_d$ = 268.1 °C, D = 10.21 km·s$^{-1}$, P = 44.45 GPa) and $\{[AG]_2(Cu(btm)_2)]\}_n$ (N% = 65.41%, $\rho$ = 1.826 g·cm$^{-3}$, $T_d$ = 212.5 °C, D = 10.97 km·s$^{-1}$, P = 53.92 GPa).

**Figure 6.** Extensive hydrogen bonding in the structure of $\{[AG]_2(Cu(btm)_2)]\}_n$ [49]. The aminoguanidinium nitrogen atoms are shown in green. Red dashed lines represent hydrogen.

From the above examples, compared with 1D and 2D EMOF materials, the energetic materials with 3D skeletons have the most coordination bonds, the most complex structure and are more stable. Therefore, 3D EMOF materials generally have higher heat resistance and lower sensitivity. On the other hand, ligands play the role of "scaffold" in the 3D skeleton structure because of the length and flexibility of ligands, which enable them to form many pores of various sizes and shapes, and thus may reduce the density of 3D EMOF materials [46,55]. In addition, in the 2D and 3D structures, sometimes the presence of water molecules will lead to the existence of strong hydrogen bonds in the structure

that will improve the density and strength of the crystal structure, but will have a negative impact on the decomposition temperature. A large number of experiments have been conducted to study the effects of different energetic ligands on the structure and properties of EMOFs. The relative molecular weight of the energetic groups on the ligands and the coordination space that may be provided during the coordination process might affect the coordination results [38].

## 3. Constructing Strategies of N-Rich EMOFs

### 3.1. Energetic Ligands

Nitrogen content is an important parameter affecting the energy level of EMOF materials. With the increase of nitrogen content, the energy level would increase correspondingly, while the stability of the material would decrease and the sensitivity would increase. In order to improve the energy density of EMOF materials, high-nitrogen-content energetic organic polymer with stable structure and multiple coordination sites, such as azide, hydrazine, imidazole, triazole, tetrazole, furazan and their derivatives [56–66], should always be selected as organic ligands, because the average bond energies of N≡N, N=N, N-N and C-N are 954, 418, 160 and 273 $kJ \cdot mol^{-1}$, respectively.

No matter what types of EMOF materials, the main ideas of constructing EMOFs are to select suitable metal ions, organic energetic ligands or to synthesize ideal organic energetic ligands through insitu synthesis and other needed methods. The selected organic ligands possess high nitrogen content, multiple coordination sites and good oxygen balance.

Azide anion ($N_3^-$) has the highest nitrogen content, at 100%, which can increase the heat of formation by about 355 $kJ \cdot mol^{-1}$ and make the decomposition products with low characteristic signal [51]. Therefore, it has attracted extensive attention in the field of EMOFs' construction. Azide anion, as a short bridging ligand group, can coordinate with metal center ions through a variety of coordination modes to construct EMOFs with different structures. One of the common strategies to construct an EMOF is to use azide anion as a ligand, together with other energetic ligands.

Nitrogen-rich heterocyclic compounds mainly include triazole, tetrazole, triazine, tetrazine, furazan and their derivatives (such as atrz, $H_2$tztr, et al.; See Figure 7 [54–76]) [77–79]. Five-membered nitrogen-rich heterocyclic compounds are traditional sources of energetic ligands in EMOFs. There are a number of nitrogen atomswith lone-pair electrons on the heterocycle that can coordinate with the metal by variable coordination modes and extending in multiple dimensions [37]. The compounds constructed with this kind of ligand have large gas generation and low characteristic signal of combustion products during decomposition. In addition, furazan-cyclic compounds have good oxygen balance, high density and heat of formation due to two N=C and two N-O bonds in the furazan group (Figure 7). However, it is difficult to coordinate with metal ion for lacking suitable coordination sites in the furazan ligand. The main idea is to synthesize the furazan derivatives with multiple coordination sites, such as 4,4'-diamino-3,3'-azofurazan (DAAzF), Bis(1-hydroxytetrazolyl) ($H_2$BOTFAZ) and 4,4'-oxybis[3,3'-(1H-5-tetrazoyl)]furazan ($H_2$BTFOF). Hydrazine ($H_2$N-N$H_2$) and its derivatives (such as hydragoic acid andhydrazinecarboxylic acid) [37] have also received extensive attention due to their bidentate coordination ability and high detonation heat. Hydrazine could produce exclusively gaseous decomposition products and hydrazine carboxylic acid could possess at least three different coordination modes. These types of EMOF materials includes CHP, CHHP, ZnHHPand so on.

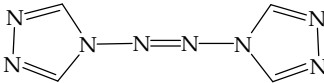

**3-amino-1,2,4-triazole; atrz**
**(N%:68.3)**

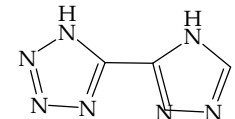

**3-(tetrazol-5-yl)triazole; H₂tztr**
**(N%:71.54)**

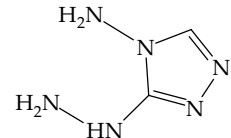

**3-hydrazino-4-amino-1,2,4-rtiazole; HATr**
**(N%: 73.65)**

**Bis(1H-tetrazol-5-yl)amine; H₂BTA**
**(N%:82.34)**

**5,5'-hydrazinebistetrazole**
**(N%:83.31)**

**Di(1H-tetrazol-5-yl)methane**
**(N%: 73.66)**

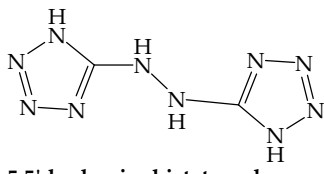

**Bi-5,5'-dinitro-2H,2H'-3,3'-bi-1,2,4-triazole;DNBT**
**(N%:68.3)**

**3,4-diaminofurazan; DAF**
**(N%:55.98)**

**4,4'-diamino-3,3'-azofurazan; DAAzF**
**(N%:57.13)**

**Bis(1-hydroxytetrazolyl)furazane;**
**H₂BOTFAZ**
**(N%:58.82)**

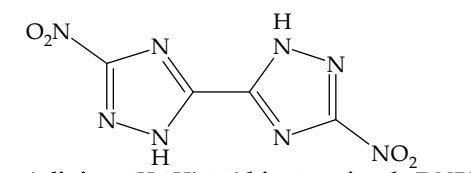

**4,4'-oxybis[3,3'-(1H-5-tetrazol)]furazan; H₂BTFOF**
**(N%: 57.93)**

**Figure 7.** Nitrogen-rich heterocyclic compounds [54–76].

### 3.2. Ion-Exchange Modification

As a novel EMOF material, energetic cation metal skeleton has been paid close attention to and reported in recent years because of its high density, high detonation heat and ordered channels. In general, the ions located in the cationic metal skeleton are inorganic anions, such as $NO_3^-$, $ClO_4^-$ and $BF_4^-$, which do not contain energy, or have low energy, and play a role to balance the charge. It is difficult to further improve and adjust the performance of energetic materials by changing the energetic ligands. However, the inorganic anion located in the skeleton structure is only combined with the cation body skeleton by relatively weak electrostatic action, which makes it possible to introduce a higher-energy anion, such as $N(NO_2)_2^-$ and $C(NO_2)_3^-$, into the skeleton structure through ion exchange. It is an effective way to produce high-energy-density materials.

In 2016 [80], a novel HEMOFs material of $\{[Cu(atrz)_3[N(NO_2)_2]_2\}_n$ $[N(NO_2)_2^-\subset MOF(Cu)]$ was reported to develop based on the feature of multiple channel of $[Cu(atrz)_3(NO_3)_2]_n$ [EMOF(Cu)]. In the 3D structure of this material, the metastable dinitramide anion $[N(NO_2)_2^-]$ is encapsulated within the molecule channel through ion exchange (Figure 8), which prevents $[N(NO_2)_2^-]$ from water molecules. The resultant inclusion complex exhibits greatly improved energy and remarkable thermal stability ($T_d$ of 221 °C, D of 6824 kJ·kg$^{-1}$, IS of 9J, ES of 2.21J). Due to the introduction of metastable groups, the safety of this material is slightly lower than that of HEMOF (Cu), but still better than that of CL-20.

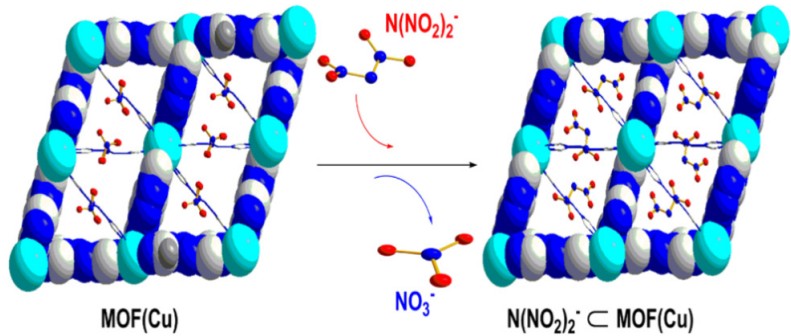

**Figure 8.** Encapsulating $[N(NO_2)_2^-]$ ion within the 3D EMOFs by ion-change process [80].

In addition, $C(NO_2)_3^-$ with a large volume also can be encapsulated within the EMOF by ion exchange with inorganic anion of three EMOF materials: $[Zn(atrz)_3(ClO_4)_2\cdot 2H_2O]_n$ (MOF(Zn)), $[Cu(atrz)_3(BF_4)_2\cdot 2H_2O]_n$ (MOF(BF$_4$)) and $[Cu(atrz)_3(NO_3)_2]_n$ (MOF(Cu)), respectively. The new corresponding materials are $\{Zn(atrz)_2[C(NO_2)_3]_2\cdot(H_2O)_2\cdot(atrz)\cdot 2H_2O\}_n$ (MOF(Zn + artz)) and $\{Cu(atrz)_2[C(NO_2)_3]_2\cdot(H_2O)_2\cdot(atrz)\cdot 2H_2O\}_n$ (MOF(Cu + artz)). Compared with other compounds containing nitroformate ions, the new materials have higher thermal stability and lower impact sensitivity and higher detonation heat than those of RDX, due to the nitrogen-based ions and the large number of hydrogen bonds in the EMOFs.

In 2018 [81], the series of new HEMOF materials of GUNI@M-BDP and FOX-12@M-BDP were reported to synthesize, whereby M is for Zn, Co or Ni.First, metal ion would be combined with 1,4-di(4'-pyrazole) benzene (BDP), to generate the compound of M-BDP. Then the energetic ions of guanidinium nitrate (GUNI) and guanylureadinitramide (FOX-12) could be filled into the structural of MOFs by two steps of ion-exchange.

### 3.3. GO-Cu(II) Modification

Many researchers have found that graphene oxide (GO) can stabilize HMX and other high EMs viaa simple method and can be used as an excellent candidate for further chemical functionalization. The metal ions can also be bound to GO sheets, forming crosslinked structures with greatlyimproved mechanical strength, as well as many other functional nanomaterials, with promising electronic properties [82]. Recently, four types of nano GO-Cu(II)-ECPs (Gozin et al., Israel) were produced

based on graphene oxide (GO)-copper(II) complex, by using 5,5'-azo-1,2,3,4-tetrazole (TEZ) and 4,4'-azo-1,2,4-triazole (atrz) as linking ligands between GO-Cu layers [83]. Depending on the reaction conditions, hermostable and insensitive ECPs or hybrid ECPs could be formed. In the preparation process, GO first reacted with $Cu(NO_3)_2$ to gain GO-Cu layers (carriers) through the hydrothermal method. Then, energy ligands of ATRZ coordinated with copper ions of carrier, introducing atrz between GO-Cu layers (Figure 9). In this series of materials, Hybrid ECP-1 has a density of 2.85 g·cm$^{-3}$, a detonation velocity of 7082 m·s$^{-1}$, an impact sensitivity greater than 98J and a friction sensitivity greater than 360N.

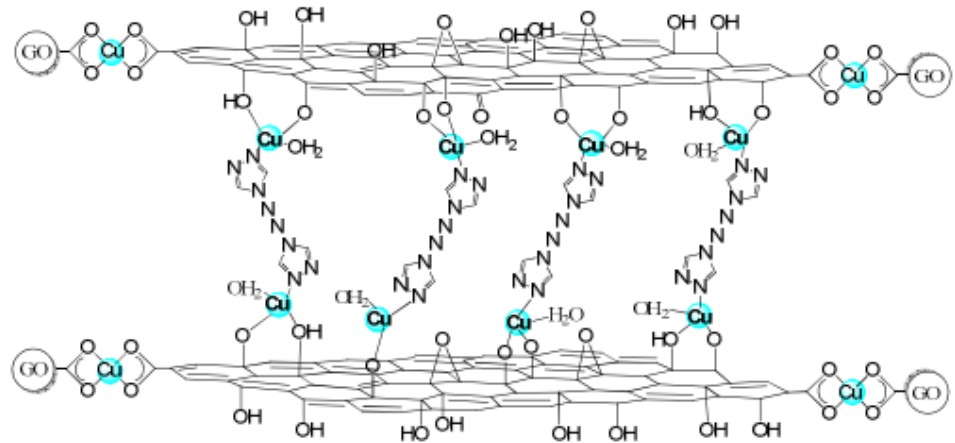

**Figure 9.** The structure of GO-Cu(II)-ATRZ ECP [83].

## 4. Applications of Nitrogen-Rich EMOFs

With stable geometric topological structure, designable energetic ligands, adjustable detonation energy performance and safety, EMOFs materials have become the research focus recently. It provides a new idea for coordinating the contradiction between high energy and low sensitivity of energetic materials. In recent years, there have been many reports on synthesis, structural characteristics, physical and chemical properties, etc., of EMOF materials, which have great application potential in the fields of green primers for the military and civilians, highly heat-resistant explosives and energetic material catalysts.

### 4.1. Green Primary Explosives

Lead azide ($Pb(N_3)_2$) and lead styphnateare the most common primary explosives. Due to the environmental pollution of lead, green primary explosives have become the important direction of the development of energetic materials, which should not only have good thermal stability, mechanical insensitivity and better performance in meeting the demands of high energy and safety, but also have environmentally friendly synthesis and detonation processes [84]. Although some non-metallic green initiation agents such as triazide melamine (CTA) and dinitrodiazopol (DDNP) have also been reported, their thermal stability and impact/friction sensitivity are increasingly unable to meet the stringent requirements of actual engineering applications (T > 180 °C [47]). Therefore, the development of green metal-based primary explosives is still the main direction, and EMOFs-based green primary explosives have shown excellent performances in this field [85]. Table 1 lists the performance parameters of lead azide, several EMOFs materials with potential as green detonators and typical elemental explosives.

**Table 1.** Physicochemical and energetic properties of some EMOF materials and classical explosives.

| Compound | $\rho$/g•cm$^{-3}$ | N/% | $T_d$/°C | $\Delta H_{det}$/kJ•g$^{-1}$ | P/GPa | D/km•s$^{-1}$ | IS (J),FS (N),ES (J) |
|---|---|---|---|---|---|---|---|
| Pb(N$_3$)$_2$ [47] | 4.800 | 28.9 | 315 | - | 33.4 | 5.877 | 2.5–4,0.1–1,- |
| {[Cu(ATZ)](ClO$_4$)$_2$}$_n$ [31] | 1.400 | 32.66 | >250 | - | - | 6.5 | 1,10,- |
| {[Zn(ATZ)$_3$](PA)$_2$·2.5H$_2$O}$_n$ [31] | 1.757 | 30.89 | 276.3 | - | - | - | 27.8,-,- |
| [Mn$_2$(HATr)$_4$(NO$_3$)$_4$·2H$_2$O]$_n$ [37] | 1.864 | 46.12 | 260 | - | - | - | -,-,- |
| [Cd$_2$(HATr)$_4$(NO$_3$)$_4$·H$_2$O]$_n$ [37] | 2.021 | 41.4 | 295 | - | - | - | -,-,- |
| CHHP [43] | 2.000 | 23.58 | 231 | 6.277 | 17.96 | 6.205 | 0.8,-,- |
| ZnHHP [43] | 2.117 | 23.61 | 293 | 6.202 | 23.58 | 7.016 | -,-,- |
| [Cu(atrz)$_3$(NO$_3$)$_2$]$_n$ [45] | 1.680 | 53.35 | 243 | 4.562 | 35.68 | 9.16 | 22.5,112,- |
| [Ag(atrz)$_{1.5}$(NO$_3$)]$_n$ [45] | 2.160 | 43.76 | 257 | 5.802 | 29.7 | 7.773 | 30,-,- |
| {[Cu(tztr)]·H$_2$O}$_n$ [44] | 2.316 | - | 80/325 | 5.524 | 31.99 | 7.920 | >40,>360,>24.7,>32, |
| [Cu(tztr)]$_n$ [44] | 2.216 | - | 360 | 14.229 | 40.02 | 8.429 | >360,>24.75 |
| K$_2$DNMAF [47] | 2.039 | 31.1 | 229 | - | 30.1 | 8.137 | 2,20,- |
| [Ag$_2$(DNMAF)·(H$_2$O)$_2$]$_n$ [86] | 2.545 | - | 230 | 8.160 | 50.01 | 9.673 | 10,160,- |
| [Ag$_2$(DNMAF)]$_n$ [86] | 2.796 | - | 212 | 9.165 | 58.30 | 10.242 | 8,120,- |
| [Pb(bta)(H$_2$O)]$_n$ [87] | 3.412 | 33.50 | 314 | - | - | - | - |
| TNT | 1.654 | 18.5 | 295 | 4.144 | 19.53 | 6.881 | 15,350,0.57 |
| HMX | 1.950 | 37.84 | 287 | 5.525 | 38.39 | 8.900 | 7.4,-,0.2 |
| RDX | 1.800 | 34.31 | 210 | 5.710 | 34.1 | 8.906 | 7.9,120,0.15 |
| CL-20 | 2.035 | 38.36 | 245 | 6.168 | 44.9 | 9.385 | 4,54,0.13 |
| PETN | 1.778 | - | 202 | 6.404 | 32.1 | 8.665 | 3,60,- |
| FOX-7 | 1.885 | - | 240 | - | 36.6 | 9.09 | -,-,- |
| TATB | 1.930 | - | 330 | 4.445 | 31.2 | 8.114 | 50,-,- |

Note: $T_d$ is decomposition temperature; $\Delta H_{det}$ is heat of detonation, calculated by EXPLO5 v6.01; P is detonationpressure; D is detonation velocity; IS means impact sensitivity; FS is friction sensitivity; ES is electrostatic sensitivity; TNT istrinitrotoluene; HMX is 1,3,5,7-tetranitro-1,3,5,7-tetrazocane; RDX is 1,3,5-trinitro-1,3,5-triazacyclohexane; CL-20 is 2,4,6,8,10,12-hexanitro-2,4,6,8,10,12-hexaazaisowurtzitane; TATB is 1,3,5-triamino-2,4,6-trinitrobenzene; FOX-7 is 1,1-diamino-2,2-dinitroethylene; PETN is pentaerythritetetranitrate.

## 4.2. Heat-Resistant Explosives

Heat-resistant explosives are a type of explosive with high temperature resistance. Currently, the most widely used high heat-resistant monomer explosives are TATB, hexanitrostilbene (HNS) and 2,6-diamino-3,5-dinitropyrazine-1-oxide (ANPyO), with the decomposition temperatures exceeding up to 30 °C. The heat-resistant explosives are mainly used for deep underground explosives and space rocket propellants, where high heat-resistance is required. However, with the harsher environment and increasing demands, it is necessary to synthesize a type of high-performance explosive with a higher heat-resistance temperature. Because of the special structural characteristics and excellent natures of EMOFs, the EMOFs-based high-resistant explosives have been widely synthesized and used.

In 2017 [88], a novel high-energy, high heat-resistant EMOFs material of potassium 4-(5-amino-3-nitro-1H-1,2,4-triazol-1-yl)-3,5-dinitropyrazole (KCPT, N% = 39, $\rho$ = 1.980 g·cm$^{-3}$, $T_d$ = 323 °C) was reported. It was synthesized via a simple one-step regioselective coupling of 3,4,5-trinitrated-1H-pyrazole (TNP), a highly energetic and the most thermally stable fully nitrated heterocyclic compound, with 5-amino-3-nitro-1H-1,2,4-triazole (ANTA), a highly heat-resistant and insensitive heterocycle in aqueous KOH. The thermal stability of this EMOF material is comparable to TATB (330 °C), and the better energetic performances of detonation heat (5.965 kJ·g$^{-1}$), detonation velocity (8.457 km·s$^{-1}$) and detonation pressure(32.5 GPa) were gained in the laboratory.

In 2017 [89], a 3D EMOF compound labeled CEMOF-1, chelating 4-amino-4H-1,2, 4-triazole-3, 5-diol (ATDO) with the divalent cations of cadmium(Cd$^{2+}$), was reported, which shows high performance of high heat resistance ($T_{dec.}$ = 445 °C), high density ($\rho$ = 2.234 g·cm$^{-3}$), good oxygen balance (−29.58%) and high activation energy. CEMOF-1 exhibits the excellent calculated detonation velocity (10.05 km·s$^{-1}$) and high detonation pressure (49.36 GPa), which is higher than those of traditional heat-resistant explosives and most 3D EMOF materials. From IS, FS and ES tests, it shows remarkably low sensitivies to external stimulus. Despite that its impact sensitivity is more than TATB and NHS, its friction sensitivity is comparable to TATB and NHS, under similar test conditions.

The material of [Zn(C$_6$H$_4$N$_5$)N$_3$]$_n$ (N% = 44.2, $\rho$ = 1.900 g·cm$^{-3}$) [41], which was recently synthesized at the university of San Diego, is the other high-performance EMOF material, with 3-pyridine tetrazolium and azide anions as the energetic ligands. It shows high heat-resistance ($T_d$ = 345 °C), and the enthalpy of combustion (−3623 kJ·mol$^{-1}$) is higher than that of HMX, TNT and RDX. However, due to the low oxygen content in the structure, its energy characteristics are relatively

low. The detonation heat, detonation velocity and detonation pressure are 1.59 kJ·g$^{-1}$, 5.96 km·s$^{-1}$ and 9.56 GPa, respectively.

## 4.3. EMOFs-Based Metastable Composites

Metastable composites (MICs) are also a research hotspot. Due to fast flame propagation speed, high energydensity and faster energy release rate, they are used as additives in solid propellant formulations and liquid fuels. MICs are generally composed of nano-aluminum (n-Al) and oxidant (such as CuO, Fe$_2$O$_3$, Bi$_2$O$_3$, PTFE, PVDF, etc.) [90]. However, it is increasingly difficult to achieve higher performance by reducing particle size alone, especially to prevent the oxidation of nano-aluminum. With the introduction of the concept of EMOFs, some novel MICs based on EMOFscame into being. Two ideas are introduced here. First, EMOF materialsare used to replace n-Al; second, EMOFs are used to wrap n-Al to prevent oxidation. The performance of the new MICs was greatly improved compared with that of the traditional MICs.

In 2018 [91], two EMOFs(Cu)-based aluminum-free MICs of 36.3% EMOFs(Cu)/63.7% NH$_4$ClO$_4$ and 43.6% EMOFs(Cu)/56.4% KClO$_4$ were reported to be synthesized. NH$_4$ClO$_4$ or KClO$_4$ would be stoichiometrically mixed with EMOFs material of [Cu (atrz)$_3$(NO$_3$)$_2$]$_n$, and followed by addingthe hexane solution and mixing ultrasonically. The MICs of EMOF(Cu)/NH$_4$ClO$_4$ or EMOF(Cu)/KClO$_4$ would be obtained by evaporating the mixing solution to remove excess hexane and water by vacuum evaporation. For EMOF(Cu)/NH$_4$ClO$_4$ and EMOF(Cu)/KClO$_4$, the IS values are 4 and 8 J, respectively, which are less than that of their aluminum-based counterparts (Al/NH$_4$ClO$_4$ and Al/KClO$_4$) and higher than that of Al/CuOnanothermite (3 J). Moreover, the FS values are 118 and 110 N, respectively, which are higher than that of their aluminum-based counterparts and nano-Al/CuO(<6N). Despite this, it was found that the frictional sensitivity of MOFs(Cu)/NH$_4$ClO$_4$ and MOFs(Cu)/KClO$_4$ is less than that of aluminum-base counterparts. More importantly, the two EMOFs(Cu)-based MICs exhibit a relatively high sensitivity to electrostatic sparks (MOFs(Cu)/NH$_4$ClO$_4$ 0.69 J, MOFs(Cu)/KClO$_4$ 0.16 J), so the operating safety margin would be wider than that of others, such as Al/NH$_4$ClO$_4$ (0.012 J), Al/KClO$_4$ (0.013 J), Al/KIO$_4$(<0.008 J), Al/NaIO$_4$(<0.008 J), Al/CuO(<0.008 J) [92], etc. The static electricity generated by human body can be up to 0.025 J [93], which can easily set off those aluminum-based thermites.

The ignition temperatures of MOF(Cu)/NH$_4$ClO$_4$ and MOF(Cu)/KClO$_4$ are 242 and 227 °C, respectively, which are much lower than that of the traditional aluminum-based MICs, and even lower than the melting point of aluminum (660 °C), mainly due to the high activity of EMOFs (Cu). The peak pressure of combustion is 6.9 MPa (MOFs(Cu)/NH$_4$ClO$_4$) and 5.7 MPa (MOF(Cu)/KClO$_4$), respectively, which is significantly higher than that of nano-aluminum-based MICs, such as Al/KIO$_4$, Al/NaIO$_4$ (4MPa), because the nitrogen-rich heterocyclic ligand and the oxygen-rich perchlorate in the structure would release a considerable amount of nitrogen gas and carbon dioxide, and a higher heat of reaction would result a higher peak pressure. Both EMOFs(Cu)-based MICs exhibit remarkably high heats of reaction, and that ofMOF(Cu)/NH$_4$ClO$_4$ (3.84 kJ·g$^{-1}$) is lower than that of nano-Al/NaS$_2$O$_8$ (6.16 kJ·g$^{-1}$) [94], but it is higher than that of most aluminum-based MICs, such as Al/CuO (1.2 kJ·g$^{-1}$) [95], Al/Fe$_2$O$_3$ (2.83 kJ·g$^{-1}$) [96] and Al/Co$_3$O$_4$ (2.50 kJ·g$^{-1}$) [97].

A multiple-level energy release system (n-Al@EMOFs) with core–shell nanostructures [98,99] can be constructed by the novel conception. First, the interfacial binding layer of polydopamine (PDA) with the center of n-Al is formed by the self-polymerization of dopamine on the surface of n-Al; then the functional groups (such as planar indole units, amino and catechol groups) in the PDA binding layer could attach to the metal ions through the coordination bonding, to form n-Al-@PDA-Cu$^{2+}$, followed by the heterogeneous nucleation and the growth of EMOFs crystals on the n-Al@PDA-Cu$^{2+}$ surfaces, to form core–shell structure. Based on the above strategies, typically, a 3D EMOFs {[Cu$_2$(DOBT)$_2$](tma)}$_n$ is successfully synthesized by solvothermal reaction of 5,5-bistetrazole-1,1-diol dehydrate (DHBT) with Cu(NO$_3$)$_2$·3H$_2$O, at 140 °C, in a mixed solvent of dimethylacetamide (DMAc) and methanol. The obtained n-Al@EMOFs shows a low ignition temperature (301.5 °C) and increased energy release (~4142 J·g$^{-1}$).

Another MIC with outstanding performance is the EF@EMOF [100] based on the same strategy. The difference is that this target core is n-Al@PVDF, an energetic nanofiber tube (EF). The obtained MIC (EF@EMOFs(DHBT)) shows increased heat release (3.464 kJ·g$^{-1}$) and burning rate (2.8 m·s$^{-1}$), as well as improved combustion efficiency. In addition, it is found that the decomposition of EMOFs and the etching reaction could generate massive gaseous products on the interface layer, which provides new channels for the further reactions and significantly improves the energy output and reaction rates.

These results highlight the great advantages over the traditional n-Al, by opening a new field for application of EMOFs, which also lays the groundwork for the development of new energetic materials. These EMOFs-based metastable compositions could be used as combustion catalysts and high energy additives for solid propellants. Further verification experiments would be required.

### 4.4. Catalysts

The oxidants of ammonium perchlorate (AP) and hexogen (RDX) are often used in the propellants and pyrotechnical formulations, which need to add an appropriate catalyst to promote the reaction. In 2006, Shreeve [101] predicted that the EMOFs' materials with bistetrazoles compounds as energetic ligands, such as *N*,*N*-bis[1H-tetrazole-5-yl]amine (H$_2$bta), could be used as catalysts in the propellants and pyrotechnical formulations. In 2011 [102], the EMOF materials of [M(BTE)(H$_2$O)$_5$]$_n$ (M = Sr, Ba; H$_2$BTE: 1,2-bis(tetrazole-5-yl)) were reported to be synthesized. They have similar crystal structures. In the 3D super-molecular structure of the EMOFs (Sr), π–π accumulation is formed between the molecular layers. The decomposition temperature curve in air atmosphere presents two stages, and the decomposition temperature in nitrogen atmosphere is greater than 350 °C. The results proved that they can be used as catalysts for AP (T$_d$ = 245 °C).

In 2013, the two 2D EMOF materials, namely Pb(Htztr)$_2$(H$_2$O)n (N% = 39.4, $\rho$ = 2.519 g·cm$^{-3}$) and [Pb(H$_2$tztr) (O)]$_n$ (N% = 27.2, $\rho$ = 3.511 g·cm$^{-3}$) [50], were reported to be synthesized. They have stable crystal structures and excellent thermal stability. The decomposition temperature would reach up to 340 and 318 °C, respectively, which is higher than that of AP (245 °C) and RDX (204 °C). The energy performances of [Pb(Htztr)$_2$(H$_2$O)]$_n$ are more satisfactory. The detonation heat, detonation velocity and detonation pressure are 5.687 kJ·g$^{-1}$, 7.715 km·s$^{-1}$ and 31.57 GPa, respectively. The impact sensitivity and friction sensitivity of the two materials are more insensitive than those of TNT, HMX, RDX and other typical explosives. It was found that both EMOF materials could promote the thermal decomposition of AP and RDX.

In 2019 [103], two novel 3D HE-MOFs materials, namely [Ag$_7$(tza)$_3$(Htza)$_2$(H$_2$tza)(H$_2$O)]$_n$ (Ma-1) and [Ag$_7$(tza)$_3$(Htza)$_2$(H$_2$tza)]$_n$ (Ma-2), were reported. Ma-1 is synthesized from the reaction of a Ag(I) ion (such as AgNO$_3$) with 1H-tetrazole-5-acetic acid (H$_2$tza) and shows the good characteristics of high energy content, excellent thermostability and insensitivity. Ma-2 is produced through the heating-dehydration of Ma-1 at 200 °C. With respect to Ma-1, Ma-2 exhibits a more excellent detonation property and almost identical safety, owing to the water-free framework. Although the decomposition temperature of Ma-2 (250 °C) is slightly lower than that of Ma-1 (263 °C), the thermostability of both compounds is comparable to that of HMX (280 °C). The two materials govern predominant detonation properties with ΔH$_{det}$ values of 8.037 and 8.455 kJ·g$^{-1}$, respectively, which are much larger than those for the traditional explosives, such as TNT, RDX, etc. The values of D, detonation velocity are 8.016 and 8.089 km·s$^{-1}$, respectively, and P, detonation pressure, are 34.54 and 35.01 GPa, respectively, for the two materials, which are comparable to that of RDX and HMX. The results of the sensitivity experiments showed that the impact sensitivity (IS) for both compounds is greater than 40 J, whereas the IS for TNT is 15 J under the similar test condition. The friction sensitivity (FS) for Ma-1 exceeds the measurable magnitude 360 N, whereas the FS value of Ma-2 is measured to be 353 N. It is observed that the exothermal temperature of AP is reduced by nearly 30 °C when AP is mixed with Ma-1 or Ma-2. The results illustrate that both compounds can behave as an effective catalyst to promote the thermal decomposition property of AP.

## 5. Conclusions and Perspective

The EMOF materials are considered as the potential candidate for HEDMs because of their excellent inherent characteristics, such as high-energy density, excellent stability, low sensitivity, etc. Especially because of the modifiability of their skeleton structure, EMOFs' materials have opened up a new way for the synthesis of various functional energetic materials (EMs). Researchers all over the world have conducted a large number of research studies on EMOF materials, including synthesis methods, structural simulation, performance tests and calculations, etc.

The traditional constructing strategies are to change the types and size of ligands that coordinate with different metal ions to form diverse EMOFs with ideal properties. Nitrogen content and oxygen balance are the most important parameters that affect the energy level of EMOFS. Five-membered nitrogen-rich heterocyclic compounds (such as triazole, tetrazole, triazine, tetrazine, furazan and their derivatives) are traditional sources of energetic ligands due to the high-nitrogen content, positive heat of formation and lone-pair electrons from the nitrogen atoms on the heterocycle. Azide, hydrazine and their derivatives anion also can be used as bridging ligands to enhance the energy level of HEMOF. Furazan ligands with high density, high positive heat of formation and good oxygen balance contain more potentialto improve the performance of explosives, which is a promising route to construct the HEMOF.

Modifying the energetic groups on the ligands, introducing high energetic anion into the skeleton and encapsulating metastable anions are promising strategies to construct more excellent EMOFs. The energetic moieties such as $NO_3^-$ and $ClO_4^-$ located in the channels can improve the stabilities and energetic properties of EMOF. On the other hand, they always limit further improvement of the detonation performance of EMOF materials. Energetic polynitro anions have significant attractive inherent characteristics, such as high densities and oxygen-rich contents, but they suffer from low stability. Non-energy and energy MOFs that are multi-porous can be used as template agents to encapsulate some energetic moieties (such as $N_3^-$, $NO_3^-$ and $N(NO_2)_2)^-$, to improve the combination performance of the whole system.

In terms of the crystal structure, the concepts of introducing energy ligands with rigid structure, connecting hydrogen bonds between groups and building $\pi$–$\pi$ stacking between layers can further improve the safety of EMOF materials and realize the integration of high energy and insensitivity for wide applications in the field of civil and military, such as the new-generation green primary explosives, the high-heat-resistant explosives and other additives of propellants and explosives. Recently, the application of nanoscale MOFs with controllable size, morphology and adjustable property has become an interesting methodology for the development novel high-energetic nanomaterials.

Although some achievements have been made, the pursuit to find high-performance and low-sensitivity EMOFsis still the eternal theme of energetic materials, guiding us to synthesize HEMOF with more excellent properties through the experiments and the theoretical calculations.

All in all, HEMOF materials, as a product of the interdisciplinary integration of structural chemistry, organic chemistry, materials science and explosive mechanics, are potential energetic materials. The development of the energetic materials should make great progress in the future. The more accurate and comprehensive synthesis, systematic measurement methods, theoretical calculation and structure simulation will be investigated in time and will eventually promote the development and application of HEMOF.

**Author Contributions:** Conceptualization, L.X. and S.X.; methodology, S.X. and X.Z.; software, J.Q.; validation, J.Q., W.G. and T.H.; formal analysis, J.Q.; investigation, W.G. and X.Z.; data curation, J.Q. and X.Z.; writing—original draft preparation, L.X.; writing—review and editing, T.H.; supervision, T.H.; project administration, S.X. and X.Z.; funding acquisition, S.X. and T.H. All authors have read and agreed to the published version of the manuscript.

**Funding:** This research received no external funding.

**Acknowledgments:** Financial support for this work was provided by the Shandong Natural Science Foundation (Nos. ZR2018MB036 and ZR2017QB009), Science Development Project of Shandong Provincial (2017GGX40115 and 2016GGX102038), Project of Shandong Province Higher Educational Science and Technology Program (J13LD08 and J17KA094) and Scientific Research Fund of University of Jinan (XBS1644).

**Conflicts of Interest:** The authors declare no conflict of interest.

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
