# Peer review of "Constructing Strategies and Applications of Nitrogen-Rich Energetic Metal–Organic Framework Materials"

_catalysts, doi:10.3390/catal10060690_

Round 1
Reviewer 1 Report
The work entitled “Constructing Strategies and Applications of Nitrogen-Riched Energetic 2 Metal-Organic Framework Materials” by Xu and co-authors aimed to provide a comprehensive review of various types of Nitrogen-rich metal–organic frameworks with potentials as energetic materials. In general, the group has developed a systematic approach in going through various types of Nitrogen-rich metal–organic frameworks ranging from 1D to 3D structures. Overall, the work is steered towards the structures of Nitrogen-rich metal–organic frameworks through the alterations of various types of chemical ligands. However, there is only a minor coverage of this class of materials for applications in catalysis and surface chemistry. Therefore the authors need to consider the following points in the revision of the manuscript to warrant publication in Catalysis:
- There was a recently published review on similar topic by Jiaheng Zhanga and Jean'ne M. Shreeve (Dalton Transactions, Issue 6, 2016). What are the new elements covered by the current manuscript compared to the published review?
- The review does not currently provide a strategic approach for designing specifically designed Nitrogen-rich metal–organic frameworks with the choice of ligand. Such coverage will lay out a rational rule in choosing nitrogen containing ligands.
- Is there any effect of synthetic approaches (e.g., one pot synthesis) in dictating the properties of Nitrogen-rich metal–organic frameworks?
Author Response
The work entitled “Constructing Strategies and Applications of Nitrogen-Riched Energetic 2 Metal-Organic Framework Materials” by Xu and co-authors aimed to provide a comprehensive review of various types of Nitrogen-rich metal–organic frameworks with potentials as energetic materials. In general, the group has developed a systematic approach in going through various types of Nitrogen-rich metal–organic frameworks ranging from 1D to 3D structures. Overall, the work is steered towards the structures of Nitrogen-rich metal–organic frameworks through the alterations of various types of chemical ligands. However, there is only a minor coverage of this class of materials for applications in catalysis and surface chemistry. Therefore the authors need to consider the following points in the revision of the manuscript to warrant publication in Catalysis:
Comment 1. There was a recently published review on similar topic by JiahengZhanga and Jean'ne M. Shreeve (Dalton Transactions, Issue 6, 2016). What are the new elements covered by the current manuscript compared to the published review?
Authors: Thanks for the reviewer’s constructive comments. The review by Jiaheng Zhang and Jean'ne M. Shreeve mainly introduced the 3D nitrogen-rich metal-organic frameworks from three aspects of azole-based 3D energetic materials, hydrothermal in situ synthesis for 3D MOFs and application. In particular, the article only provided an overview of its application. In our current manuscript, two popular classification methods of these materials have been summarized. The EMOFs materials also are classified into three types of neutral, cationic and anion EMOFs according to the charge of the main body skeleton. The strategy to improve the energy of EMOF materials by ion exchange of energetic ions instead of inorganic anions within the skeleton is also discussed. On the other hand, in part 4 of our manuscript the application prospects of these novel materials are also discussed.
Comment 2: The review does not currently provide a strategic approach for designing specifically designed Nitrogen-rich metal–organic frameworks with the choices of ligands. Such coverage will lay out a rational rule in choosing nitrogen containing ligands.
Author: Thanks for the reviewer’s constructive comments. The selected energetic ligands should have high-nitrogen content, multiple coordination sites and good oxygen balance. Nitrogen-rich azole-ring compounds are usually selected as energetic ligands because of their high-nitrogen content, positive heat of formation and lone pair electrons from the nitrogen atoms on the heterocycle. On the other hand, the selected energetic ligands would depend on the target materials and synthesis conditions. There are reviews in line 193, 210-223.
Line 193: No matter what types of EMOFs materials, the main ideas of constructing EMOFs are to select suitable metal ions, organic energetic ligands or to synthesize ideal organic energetic ligands through in-situ synthesis and other needed methods.
Line 219-224: The main idea is to synthesize the furazan derivatives with multiple coordination sites, such as DAAzF, H2BOTFAZ, H2BTFOF. Hydrazine (H2N-NH2) and its derivatives (such as hydragoic acid, hydrazinecarboxylic acid)[37] have also received extensive attention due to their bidentate coordination ability and high detonation heat. Hydrazine could produce exclusively gaseous decomposition products and hydrazine carboxylic acid could possess at least three different coordination modes.
Comment 3: Is there any effect of synthetic approaches (e.g., one pot synthesis) in dictating the properties of Nitrogen-rich metal–organic frameworks?
Author: Thanks for the reviewer’s constructive comments on our manuscript. There are flexible and various coordination models between metal ion or metal clusters and energetic organic ligands due to the multiple coordination sites on the nitrogen-rich ligands. So, under the similar or identical reaction condition, pure EMOF compound or EMOF mixtures with different structures and different ratios may be formed in the process of one-pot synthesis. Even structurally similar EMOF materials may have different physicochemical properties. In addition, the structural frameworks and dimensionalities of target MOFs could be affected by other facors, such as temperature, solvent effect, pH, templates, stoichiometry and guest molecule.
Reviewer 2 Report
This review by Xu et al. provides a nice discussion of energetic coordination polymers, their structural classes and performance, and brings together a good amount of data on these materials which will be of use to those working in the field of energetic materials. I don't have any major issues with the paper itself (except those comments which I lay out below, most of which are minor), but I am puzzled as to why this was submitted to a journal called Catalysis, when really this paper has nothing to do with catalysis, and only has a very brief mention of some catalysis accompanying detonation in one paragraph towards the end. This paper could probably be accepted following minor revisions if the editors really think this is an appropriate submission for Catalysis, but really it should be transferred to a journal like Materials or Crystals which would be much more appropriate.
Corrections:
- The authors should familiarise themselves with the IUPAC definition of a MOF, as most of the materials described here are not MOFs as they lack porosity. IUPAC defines a MOF as "A Metal-Organic Framework, abbreviated to MOF, is a Coordination Network with organic. ligands containing potential voids." A large portion of the materials described here have no (potential) porosity and would be better described as coordination polymers.
- Figure 7 really needs to be re-drawn. There are numerous errors including at least 3 different fonts, a nitrogen atom with 4 bonds and an incorrect sp3 carbon atom in the structure of DNBT, and in many places the letter N has been placed on top of a carbon atom in the framework structures.
- The English requires extensive editing (which I will leave to those who get paid to do this), but I will point out two particular issues - the repeated use of the term "double toothed" or similar must be replaced with "bidentate", and on line 283 styphnate has been mis-spelled as stephenate
Author Response
This review by Xu et al. provides a nice discussion of energetic coordination polymers, their structural classes and performance, and brings together a good amount of data on these materials which will be of use to those working in the field of energetic materials. I don't have any major issues with the paper itself (except those comments which I lay out below, most of which are minor), but I am puzzled as to why this was submitted to a journal called Catalysis, when really this paper has nothing to do with catalysis, and only has a very brief mention of some catalysis accompanying detonation in one paragraph towards the end. This paper could probably be accepted following minor revisions if the editors really think this is an appropriate submission for Catalysis, but really it should be transferred to a journal like Materials or Crystals which would be much more appropriate.
The authors should familiarize themselves with the IUPAC definition of a MOF, as most of the materials described here are not MOFs as they lack porosity. IUPAC defines a MOF as "A Metal-Organic Framework, abbreviated to MOF, is a Coordination Network with organic ligands containing potential voids." A large portion of the materials described here have no (potential) porosity and would be better described as coordination polymers.
Author: Thanks for the reviewer’s constructive comments on our manuscript. The definition of EMOF materials is based on that of MOF because the construction concept of EMOF arises from the concept of MOF. MOFs materials have attracted extensive interest because of their stable architectures, controllable structures, modifiable properties, which could be benefit to design low sensitive materials. The EMOF materials are results of requirements of energetic materials with high energy and low sensitivity combining with the inherent characteristics of MOF. To improve energetic level of energetic materials, the organic energetic ligands are be selected during the process of constructing metal-organic frameworks. The researchers are more concerned with density and stability. The multi-porosity would reduce the density of materials, which would lead to low energetic level.
Comment 1. Figure 7 really needs to be re-drawn. There are numerous errors including at least 3 different fonts, a nitrogen atom with 4 bonds and an incorrect sp3 carbon atom in the structure of DNBT, and in many places the letter N has been placed on top of a carbon atom in the framework structures.
Author: Thanks for the reviewer’s careful examination on our manuscript. Now, we have redrawn the Figure 7 in Line 203 to 209.
Figure 7. Nitrogen rich heterocyclic compounds.[54-76]
Comment 2. The English requires extensive editing (which I will leave to those who get paid to do this), but I will point out two particular issues - the repeated use of the term "double toothed" or similar must be replaced with "bidentate", and on line 283 styphnate has been mis-spelled as stephenate.
Author: Thanks for the reviewer’s careful examination on our manuscript. Now, we have revised the English.
Except the above listed revisions based on the reviewers’ comments, we also supplemented references. Now, the whole number of the references reaches up to 103. On the other hand, we also added some other contents in the text.
Line 394-395: These EMOFs-based metastale compositions could be used as combustion catalysts and high energy additives for solid propellants. Further verification experiments should be required.
Line 443-447:Nitrogen content and oxygen balance are the most important parameters that affect the energy level of EMOFS. Five-membered nitrogen-rich heterocyclic compounds (such as triazole, tetrazole, triazine, tetrazine, furazan and their derivatives) are traditional sources of energetic ligands due to the high-nitrogen content, positive heat of formation and lone pair electrons from the nitrogen atoms on the heterocycle.
